# Congenital Heart Diseases: Recent Insights into Epigenetic Mechanisms

**DOI:** 10.3390/cells14110820

**Published:** 2025-05-31

**Authors:** José Manuel Rodríguez-Pérez, Diego B. Ortega-Zhindón, Clara Villamil-Castañeda, Javier Santiago Lara-Ortiz, Verónica Marusa Borgonio-Cuadra, Jorge L. Cervantes-Salazar, Juan Calderón-Colmenero, Zeomara Nathali Escalante-Ruiz, Eduardo Retama-Méndez, Yessica C. Hernández-García, Nonanzit Pérez-Hernández

**Affiliations:** 1Department of Molecular Biology, Instituto Nacional de Cardiología Ignacio Chávez, Mexico City 14080, Mexico; josemanuel_rodriguezperez@yahoo.com.mx (J.M.R.-P.); clara.vi.ca1@gmail.com (C.V.-C.); santibuho25@gmail.com (J.S.L.-O.); zeoruiiz1@gmail.com (Z.N.E.-R.); zemado@hotmail.com (E.R.-M.); 2Department of Pediatric Cardiac Surgery and Congenital Heart Disease, Instituto Nacional de Cardiología Ignacio Chávez, Mexico City 14080, Mexico; diegob.ortegaz@gmail.com (D.B.O.-Z.); jorgeluis.cervantes@gmail.com (J.L.C.-S.); charytin.med@gmail.com (Y.C.H.-G.); 3Programa de Maestría y Doctorado en Ciencias Médicas, Odontológicas y de la Salud, Universidad Nacional Autónoma de México, Mexico City 04510, Mexico; 4Programa de Doctorado en Ciencias Biomédicas, Universidad Nacional Autónoma de México, Mexico City 04510, Mexico; 5Laboratory of Genomic Medicine, Department of Genetics, Instituto Nacional de Rehabilitación Luis Guillermo Ibarra Ibarra, Mexico City 14389, Mexico; vborgoni@yahoo.com.mx; 6Department of Pediatric Cardiology, Instituto Nacional de Cardiología Ignacio Chávez, Mexico City 14080, Mexico; juanecalderon@yahoo.com.mx; 7Academic Division of Health Sciences, Universidad Juárez Autónoma de Tabasco, Villahermosa 86100, Tabasco, Mexico

**Keywords:** congenital heart diseases, epigenetics, DNA methylation, histone modifications, polycomb and trithorax protein complexes, non-coding RNAs

## Abstract

Congenital Heart Diseases (CHDs) are a heterogeneous group of structural abnormalities affecting the heart and major arteries, which are present in at least 1% of births worldwide. Studies have linked CHD to both genetic and environmental factors. In this regard, it has been demonstrated that changes in the epigenetic pattern impact the expression of key genes involved in proper cardiac development. Therefore, it is suggested that aberrant epigenetic mechanisms may contribute to the development of these pathologies. Here, we review and summarize the main epigenetic mechanisms involved in CHD. Moreover, cardiac development and the importance of the environment and CHD are also addressed.

## 1. Introduction

Epigenetics is the field that studies the structural modifications that occur in the genome and modulate gene expression without altering the nucleotide sequence. These modifications are driven by diverse mechanisms, among which DNA methylation, histone modifications, and non-coding RNAs stand out. All are key in regulating gene expression and accessibility [1].

In this framework, alterations of epigenetic modifications impact several cardiac structures, promoting the manifestation of a wide range of irregular phenotypes, such as Congenital Heart Diseases (CHDs). This term encompasses a set of heterogeneous structural anomalies that jeopardize the heart and the major arteries [2]. These cardiac structural alterations are present in at least 1% of births worldwide [3], with the highest incidence reported in Europe and North America (9.4 per 1000 live births) [4]. The classification of CHD has been performed in several ways in the past. The report of the guidelines for the management of adults (AHA/ACC 2018) with Congenital Heart Disease does so according to their anatomy and physiological complexity [5]. From this point of view, simple or non-complex defects include those that can be treated without surgery, such as ventricular and atrial septal defects (VSDs, ASDs), respectively, or defects like Patent Ductus Arteriosus (PDA); meanwhile, complex and critical defects require immediate medical attention and surgery during the first year of life, as is the case with Tetralogy of Fallot (TOF), Ebstein anomaly (EA) or Hypoplastic Left Heart Syndrome (HLHS) [6]. From a clinical and often more convenient standpoint, these pathologies can also be divided between cyanotic and non-cyanotic. The first kind are characterized by a right-to-left shunt resulting in cyanosis and include conditions such as TOF, Tricuspid Atresia (TA) and transposition of the great arteries (TGA), while the second kind can be obstructive, like Pulmonary Stenosis (PS) or Coarctation of the Aorta (CoA) [7].

It has been shown that changes in epigenetic signatures have a direct impact on key genes involved in heart development. For example, hypermethylation of the *TBX20* gene has been identified in patients with TOF, ASD, and VSD, suggesting an anomalous epigenetic regulation that may contribute to the development of these pathologies [8,9]. Likewise, non-coding RNAs take on crucial roles in heart development, as seen, for example, with the downregulation of miR-1, which is associated with impaired differentiation of embryonic stem cells (ESCs) towards cardiomyocytes, highlighting the importance of microRNAs (miRNAs) for the adequate formation of cardiac tissue [10]. This evidence supports a role for epigenetic modifications as determinants in the occurrence and progression of CHD, directly impacting key cardiac anatomic structures, such as the heart valves, the atrial or ventricular walls, and the major arteries [11].

Intrinsically, cardiac embryology is determined by interactions with the environment, mainly through hemodynamic signaling, which leaves the embryo vulnerable to anomalous development in response to external hemodynamic stimuli but also gives it a high degree of adaptability, which explains the limited spectrum of viable phenotypes up until birth. The multiple and complex levels of gene regulation make it difficult to unravel the precise epigenetic mechanisms involved in CHD.

Currently, the use of genomic technologies that allow for the identification of a wide range of genetic defects, like Third-Generation Sequencing (TGS) and Genome-Wide Association Studies (GWASs), has allowed for significant progress in the field of genomic medicine. Particularly, TGS involves wide and high-quality reads, permitting the better comprehension of genomic structure and avoiding previous amplification by PCR. This technique has already been applied in several studies involving genetic diseases, and recently it has been applied to genome assembly, structural variant detection, epigenetic analysis, and pharmacogenomics (PGx) profiling studies [12,13].

Furthermore, GWASs represent a fundamental tool for identifying genetic variants associated with complex diseases and other relevant phenotypes. These studies include wide arrays of individuals. At the moment, the use of GWASs has positively impacted research into multifactorial illnesses, identifying new robust associations of several alleles influencing disease phenotypes in a highly reproducible manner [14,15]. The arrival of these tools has therefore enhanced research into the genetics and epigenetics of CHDs, making them essential for advancing research in this field.

Therefore, the early detection of epigenetic marks associated with CHD is crucial for better prevention and the possibility of personalized therapeutic approaches that benefit patients. Here, we review and summarize the role of the epigenetic mechanisms involved in CHD.

## 2. Epigenetics

The term “epigenetics” was first used in 1942 as a way to regard the regulation of gene expression without altering the nucleotide sequence, implying the control of transcription and translation. Additionally, it was determined that cell differentiation depends on the activity or inactivity of certain genes [4]. Then, it was established that epigenetic mechanisms are conditioned by environmental influences and that they regulate gene expression while being kept during cell division, and the term “epigenetic control of gene expression” was coined. The first epigenetic mechanism described was DNA methylation. Although this modification had been known about since the 1940s, its relationship with epigenetics was described thirty years later [16]. Afterwards, other epigenetic mechanisms were described, such as histone modifications, which control the accessibility of gene regulatory regions, and non-coding RNAs, which can control mRNA translation [17,18].

The regulation and maintenance of cell-type-specific gene expression patterns is one of the main functions of epigenetic modifications [16]. The ability of cells to stably retain and transmit unique expression patterns to their daughter cells is known as epigenetic memory, and it is coded by epigenetic marks. This not only defines the characteristics of germ cells and differentiated cells but also the inherited properties of embryonic cells and their line, as well as cells that define a disease state [19].

During embryonic development, epigenetic memory plays a crucial role in cell differentiation. Stem cells and precursor cells use epigenetic marks to remember their identity and function, ensuring their differentiation towards specific cell types and contribution to the adequate formation of tissues and organs. Inherited epigenetic modifications guarantee that progenitor cells are kept in their differentiated state throughout development [19].

The advent of omic technologies and high-throughput techniques in biomedical research has contributed to new tools for studying genome-wide epigenetic marks, resulting in the field of epigenomics. For instance, the ability to analyze the tri-dimensional organization of chromatin in the cell nucleus offers a profound vision of genome organization. On the other hand, unicellular epigenomics provides information on epigenetic heterogeneity among cells, which harbors great potential for the diagnosis and prognosis of cardiovascular diseases (CVDs). In turn, genome sequencing after treatment with sodium bisulfite represents the standard for the detection of methylated cytosine bases, while sequencing coupled to immunoprecipitation has been adopted for the study of epigenetic modifications at a large scale as other next-generation sequencing techniques continue to be adapted for epigenomic studies [20,21]. Through these techniques, it has been shown that modifications in DNA methylation regulate the gene expression of cardiomyocytes during fetal development and that alterations in these processes can contribute to the development of CHD.

## 3. Congenital Heart Diseases and the Environment

Throughout their lifetimes, human beings are exposed to different environments, which can result in the exposure to pollutants or substances that drive us towards disease states.

Environmental expositions are linked to the pathogenesis of many illnesses through multiple molecular pathways that affect the epigenetic regulators of gene expression. These modifications reflect the cumulative environmental exposures across a lifetime that may, on occasion, lead to disease development. Besides providing a link between environmental stress factors and disease pathogenesis, epigenetic modifications can work as trustworthy markers of subclinical disease and even accelerated aging [22]. Although it is challenging to accurately measure the level of exposure and associated risk for pregnant women and their children, several pollutants like gases, pesticide subproducts, and heavy metals have been linked with a higher risk of CHD (Table 1) [23]. Many human diseases result from the interaction between the genome and the surrounding environment; these gene–environment interactions show how genes and environmental factors affect human features. Environmental exposure may increase susceptibility to illnesses through modifications to epigenetic marks [24].

Fetal exposure to toxic substances is an example of a potential cause of epigenetic deregulation since many of these compounds alter gene expression, resulting in phenotypes associated with congenital heart malformations [37]. Among the many drugs known to be teratogens, some have been found to affect heart development in a direct and specific manner (Table 1) [38].

One known example is thalidomide, which was used as an antiemetic for pregnant women and has been shown to influence cardiogenesis. Interestingly, it induces a defect similar to that observed in Holt–Oram syndrome, caused by a mutation on *TBX5* and associated with CHDs, mainly VSD, as well as limb abnormalities [39].

Furthermore, trichloroethene, a halogenated hydrocarbon present in polluted water sources, increases the risk of heart malformations by decreasing the expression of nitrous oxide synthase (NOS), which exposes the embryo to higher levels of free radicals. Moreover, the epithelial–mesenchymal transition of valve progenitor cells is altered, and vascular endothelial growth factor (VEGF) signaling is affected [23].

Dioxin exposure in pregnant women has been associated with an increase in the incidence of congenital heart malformations. A study in Baltimore, USA, identified a group of hypoplastic left heart syndromes related to this compound. The authors found dioxin to directly impact cardiomyocyte differentiation by altering genome methylation due to a decrease in regulation by DNA methyltransferases DNMT3A and DNMT3B [40]. These examples highlight the importance of simultaneously evaluating genetic and environmental factors to identify high-risk genotypes and vulnerable populations [24].

## 4. Cardiac Development

Heart development is a highly complex process; it is the first organ to develop during embryogenesis, and its development starts during gastrulation in the second week of human development [41]. This organ, with noteworthy asymmetry, is derived from four different sources of progenitor cells: the extraembryonic mesoderm, the embryonic mesoderm, cardiac progenitor cells, and neural crest cells [42]. 

Both during development and after birth, heart function and size are closely related; it must be big enough to satisfy its physiological functions but not big enough to provoke a circulatory obstruction. Contractile cells, known as cardiomyocytes, constitute the majority of cardiac tissue and perform a crucial role in normal cell hypertrophy, which is essential for normal heart growth after birth [43]. In the neonatal stage, cardiomyocytes stop proliferating and experience hypertrophic growth, increasing their diameter and mass. As the heart grows, an essential metabolic change occurs, passing from an anaerobic state dependent on glycolysis to an oxidative metabolism, which influences the postnatal proliferation, differentiation, and maturation of cardiomyocytes. This change is a characteristic seal of the transition from the fetal to the adult stage [43].

Epigenetics performs a crucial role in the regulation of genes, controlling progenitor cell differentiation, and ensuring adequate heart development. DNA methylation and histone modifications are fundamental mechanisms in this process. The complex process of cardiogenesis starts during gastrulation and continues with the transformation of the embryoblast into three germ layers: the ectoderm, mesoderm, and endoderm. The heart is formed from mesoderm cells that migrate towards the anterolateral border of the trilaminar embryonic disk. These cells are differentiated into cardiomyocytes under the influence of factors like Bone Morphogenic Protein (BMP), as well as the essential transcription factors ISLET1 and NKX2.5 [44,45]. During the third week of development, the heart acquires a three-dimensional shape due to the rapid growth of neuronal tissue. Cell migration and gene expression regulation are essential to this process, and any alteration can lead to congenital defects like ectopia cordis [46,47].

From the fourth week of development, the cardiac straight tube goes through a looping or bending process. This process involves the rupture of the dorsal mesocardium across its middle line, provoking a rightwards tilting of the cardiac tube, which acquires a “C”-like shape. While the tube ends continue growing linearly, the center of it expands radially, forming a unique, outward-projecting ventricle. In this stage, four cardiac regions can be distinguished: the atrium, atrioventricular channel (AVC), ventricle, and outflow tract (OFT). As development progresses, the cardiac tube’s curvature becomes more complex, adopting an “S”-like shape [46,48].

During bending, the cardiac tube’s length increases by five times due to the continuous incorporation of differentiated cardiomyocytes, which come from a group of rapidly proliferating mesoderm cells localized in the heart venous region; the high rate of proliferation of these cells, also known as second heart field progenitors, is regulated by canonical Wnt/β-catenin signaling. Second heart field progenitors express the transcription factor ISLET1 while they incorporate into the cardiac tube, and once differentiated into cardiomyocytes, these cells cease proliferating, with a corresponding decrease in ISLET1 and increase in NKX2.5. Another important transcription factor, TBX1, is known to be a key regulator of second heart field cell segregation towards the entry and exit poles of the heart [49].

It is important to highlight that an alteration of left–right asymmetry during this stage of heart development can result in complex disease phenotypes, like heterotaxy syndrome, transposition of the great arteries (TGA), double-outlet right ventricle (DORV), and double-outlet left ventricle (DOLV) [50,51,52]. Although several hypotheses have been proposed to explain the mechanisms that start and maintain this asymmetry in heart development, these are not yet fully understood [53].

Heart bending and chamber specification are completed on approximately the twenty-eighth day of embryonic development, or at the fourth week of gestation. Nevertheless, the cardiac organ does not acquire its final form until approximately the 50th day (seventh week of gestation). During this period, three key morphological events occur: (1) septation, which divides the four heart chambers and splits the arterial trunk into the aorta and the pulmonary artery; (2) the formation and fusion of the pulmonary veins and vena cava with the left and right atriums, respectively; (3) the formation of the four heart valves [46]. Afterwards, muscular septums emerge from the atrium ceiling and the ventricle floor, growing towards the atrioventricular cushions. The division of the arterial trunk into the aorta and pulmonary artery is greatly attributed to the population of cardiac progenitors that migrate from the neural crest [49].

Most complex CHD cases develop during these last three morphological stages, since an incorrect chamber septation may result in ASD or VSD, which may occur in an isolated manner or alongside other malformations, like TOF. Incorrect septation of the arterial trunk can cause DORV or DOLV, TGA, PDA, and CoA. On the other hand, when the pulmonary veins are not adequately fused with the left atrium, an “anomalous pulmonary venous return” is produced, where the veins flow towards the right atrium instead of the left. Furthermore, valve malformations can include their total absence (as in Tricuspid Atresia), narrowed valves (as in Pulmonary Stenosis), and anomalous valve geometries (as in bicuspid aortic valves). Malformed septums, valves, and blood vessels can significantly alter the mechanical environment of the adjacent chambers during heart maturation, which may lead to pathologic remodeling and syndromic effects [54].

The heart and placenta share developmental pathways and are tightly related from the first weeks of gestation. Placental anomalies can be related to congenital heart defects. The anomalous insertion of the umbilical cord and placental weight reduction are frequently found in fetuses with CHD, mainly with TOF, DORV, and severe VSD. The disturbance of the transcription factor HAND1 has been found to affect both placental and cardiac development, provoking an anomalous vascular remodeling and the lessened thickness of the heart wall [55]. In studies with mice, HAND1 disruption has been found to increase fetal mortality and affect the heart structure [56]. Moreover, feto-fetal transfusion syndrome in monochorionic twins has been associated with a higher risk of CHD. These twins have been observed to have a nine-times-higher risk of developing CHD compared to unique births, owing to their abnormal placentation and the deregulation of angiogenic factors. It has been proposed that hypoxia derived from deficient maternal–fetal circulation can explain the higher prevalence of CHD in monozygotic twins [57], as it provokes a reduction second heart field progenitor cell proliferation, which interferes with the expression of transcription factor NKX2.5, which in turn affects the development of the outflow tract, the atriums, and the right ventricle. This can result in defects like ToGA, overriding aorta, DORV, VSD, and PDA [58,59].

## 5. DNA Methylation

DNA methylation is one of the most studied epigenetic marks involved in genome regulation. Molecularly, the process involves the addition of a methyl group on the carbon 5 of cytosine, resulting in 5-methylcytosine (5mC) and a structural alteration [1,16,60]. In mammals, this process commonly occurs at CpG sites, nucleotide pairs of a guanine followed by a cytosine, which are methylated in most of the genome but can also be found grouped in genomic regions known as CpG islands, which are often demethylated. Most gene promoters are contained within these islands, favoring the binding of transcription factors to these sites and transcriptional activation, hence the reason why a change in methylation state produces changes in gene expression [61]. During the different developmental stages, the remodeling of methylation patterns in the genome occurs, which is crucial for the embryo formation processes, including cardiogenesis, for which a precise gene expression in time and space is essential [62,63]. The main actors in this process are DNA methyltransferases (DNMTs), which are in charge of performing DNA methylation, and ten-eleven translocation enzymes (TETs), which perform demethylation [64]. Some studies suggest a decrease in de novo methyltransferase expression, particularly DNMT3A and DNMT3B, as a key factor in CHD pathogenesis [65]. Nevertheless, no specific mechanisms have been identified that explain this phenomenon. Research on this process has associated deregulated methylation with a number of pathologies, including CHD.

Previously, methylation states in CHD patients have been addressed. For instance, a study of genome-wide DNA methylation in a cohort of 24 newborns who had aortic valve stenosis (AVS) compared with gestational-age-matched controls identified significantly altered CpG methylation at 59 sites in 52 genes. Furthermore, Gene Ontology analysis identified biological processes and functions for these genes including the positive regulation of receptor-mediated endocytosis, and the analysis of molecular function categories, as determined using the Database for Annotation Visualization and Integrated Discovery, identified low-density lipoprotein receptor binding, lipoprotein receptor binding, and identical protein binding to be over-represented in the AVS group. A significant epigenetic change in *APOA5* and *PCSK9* was highlighted, as these genes are known to be involved in AVS. Significantly, a large number of CpG methylation sites individually demonstrated good to excellent diagnostic accuracy for the prediction of AVS status, thus raising the possibility of having biomarkers for this disorder [66]. However, no functional studies were performed in this study to enhance the findings.

Tetralogy of Fallot is one of the most common and complex CHDs, characterized by four main cardiac anomalies that affect heart structure and function [67,68]. Zhu et al. performed an in vitro observational study in patients with TOF, where they observed a decreased *NOTCH4* expression in TOF patients compared to healthy controls, leading them to analyze the methylation status of the *NOTCH4* gene promoter region by bisulfite pyrosequencing. The authors found a hypermethylation state at the CpG 2 site, and found that *NOTCH4* expression was negatively associated with CpG2-site methylation. Further investigation into the mechanism which controls this process led them to uncover a resulting decrease in the affinity of the *NOTCH4* promoter with the ETS-1 transcription factor following in vitro methylation, as seen via dual luciferase reporter assays and electrophoretic mobility shift assays, resulting in a decrease in NOTCH4 expression and possibly contributing to the development of TOF [69].

Notch signaling is known to be involved in several cellular processes, like proliferation, cell death, and cell fate decisions. Thus, this pathway is involved in the development of most organs, including in cardiogenesis. During heart development, notch signaling continuously functions throughout the processes of cardiac specification, endocardium patterning, valve and chamber morphogenesis, and even cardiac regeneration, often also interacting with other relevant developmental pathways like WNT and BMP. Therefore, it stands to reason that differential expression of *NOTCH* genes like *NOTCH4* could be a decisive factor in the development of CHD [70].

On the other hand, Xiaodi et al. assessed the methylation state of nuclear receptor subfamily 2 group F member 2 (NR2F2). This *NR2F2* gene encodes a ligand-inducible transcription factor involved in angiogenesis and heart development. These researchers aimed to elucidate the molecular mechanism of the epigenetic regulation of *NR2F2* in TOF development in tissues from patients with TOF compared with healthy controls. The authors found the CpG island shore (CGIS) in the promoter to be hypomethylated at the CpG 3 site, and observed this to be a differentially methylated region with a significant negative correlation with NR2F2 expression in TOF patients’ tissue. Furthermore, through functional analysis, the authors reported the nuclear receptor RXRɑ to regulate NR2F2 expression, and suggested that promoter hypomethylation increases its affinity with this factor, thereby raising gene expression, potentially having a role in TOF [71].

Recently, Zhou et al. analyzed global methylation state. The authors used myocardial tissue from fetuses diagnosed with cardiac defects to identify global changes in methylation through microarrays, as well as the methylation state at CpG sites in the vicinity of genes involved in cardiogenesis. The authors reported global hypomethylation in the CHD fetuses compared to healthy fetuses. Moreover, they found differentially methylated regions distributed across the genome and not only in promoters. An enrichment functional analysis was also performed, and it revealed hypomethylated genes to be involved in cardiac tube morphogenesis and immune regulatory functions. Additionally, the three most significant differentially methylated regions involved in cardiogenesis were studied, corresponding to the *EGFR*, *NOTCH1*, and *SL1C19A* genes, and the researchers found an important decrease in expression at the protein level [72].

One group of relevant genes is those in the T-box transcription factor family, which are involved in cell fate decisions, morphogenesis, and organogenesis during heart development [73]. Yang et al. elucidated the CpG island and transcription factor binding to the *TBX20* gene promoter through a bioinformatic analysis. The authors also assessed the promoter’s methylation state and gene expression in TOF patients, where they found hypomethylation compared to their controls, including at the predicted CpG sites, as well as an increase in *TBX20* mRNA. The authors suggest that the disrupted methylation pattern could affect transcription factor TFAP affinity, which has a suppressive activity against *TBX20*, and, therefore, that the altered methylation results in gene overexpression, which may contribute to TOF pathogenesis [9].

Furthermore, Gong et al. analyzed *TBX20* gene promoter methylation by bisulfite sequencing. These authors reported hypomethylation in TOF patients compared to their control and confirmed the impact of *TBX20* on transcriptional activity by luciferase assay, meaning that methylation changes within this region are responsible for modulating this activity. Furthermore, the authors report that the transcription factor Sp1 is able to bind to this genomic region, and that this process is inhibited by the methylation of the binding sites, as confirmed by the luciferase assay. These findings support a possible role of the epigenetic regulation of *TBX20* in modulating other transcriptional programs which could well be involved in TOF pathogenesis [8].

Recently, García-Flores et al. assessed CpG sites in the *TBX5* and *TBX20* gene promoters using peripheral blood from pediatric patients presenting septal defects through pyrosequencing; by analyzing these genes, the authors found hypermethylation, on average, at the evaluated CpG sites in the patient group compared to individuals with PDA as their control group. Moreover, through a Receiver Operating Characteristic (ROC) analysis, they showed that the observed hypermethylation can be used as a risk marker for presenting septal defects. In both genes, the association of the methylation levels with the environmental factors that the mothers were exposed to was analyzed, suggesting a protective factor associated with vitamin consumption for the *TBX20* gene and a risk association with infections during pregnancy for both genes [9,74].

Furthermore, Mouat et al. reported a global genome methylation analysis of dried blood samples from newborns with CHD and Down Syndrome, in which they evaluated the neonatal methylation state of those with a CHD diagnosis and compared it with that of newborns without CHD. The authors found a global hypermethylation profile in male CHD-DS patients, and also observed a sex-specific methylation signature [75].

Moreover, DNA methylation has a great influence on the establishment of imprint patterns, and some disorders have been associated with various methylation modifications in the imprinting control regions. Epigenetic imprinting is particularly vulnerable in early embryonic development, which is relevant to CHD. In a recent study, where the alteration of DNA methylation status was considered to be one of the key factors in CHD emergence and development, methylation alterations were identified in the regional differential germinal methylation regions of eight imprinted cardiac genes in CHD patients. The methylation status of these genes varied according to CHD heterogeneity; nevertheless, six of them (*GRB10*, *PEG10*, *INPP5F*, *PLAGL1*, *NESP,* and *MEG3*) were associated with a significantly higher risk of CHD. These findings suggest that an alteration in imprinted gene methylation could impact CHD development, although additional experimental studies are required to explore the causal relation between alteration in the methylation status of this group of genes and specific pathologies [76].

There are few reports on gene methylation with functional studies that allow for the assessment of causality in CHD.

## 6. Histone Modification

DNA is packed in the condensed chromatin structure found in the eukaryotic cell nucleus. Chromatin is packed in nucleosome subunits, made up of protein octamers comprising two copies of each of the nuclear histones H2A, H2B, H3, and H4 wrapped by a DNA strand [77]. Histone modifications like phosphorylation, ADP ribosylation, ubiquitination, citrullination, SUMOylation, methylation, and acetylation impact the chromatin structure, regulating the access of transcription factors or initiation complexes into genomic regions and modulating gene expression patterns in the cell [23]. Histone methylation and acetylation are the most studied modifications.

The patterns of histone methylation provide specialized binding surfaces that attract protein complexes responsible for chromatin remodeling and transcriptional regulation. This modification is regulated by enzymes like lysine methyltransferases (KMTs), which catalyze the transfer of methyl group to lysine residues, and lysine demethylases (KDMs), which eliminate methylation marks from lysine residues. Methylation in the lysine residues H3K9, H3K27, and H4K20 is associated with transcriptionally inactive heterochromatin regions, while H3K4, H3K36, and H3K79 methylation is associated with transcriptionally active euchromatin [78]. On the other hand, histone acetylation is associated with the activation of gene expression as it unpacks chromatin. The acetylation process is carried out by histone acetyltransferases (HATs) [79] and can occur in several lysine residues in histones, like H3K27, which is frequently found in transcriptionally active regions. At the same time, the opposed histone deacetylation mechanism is catalyzed by histone deacetylases (HDACs) [80].

Histone modifications intervene in the epigenetic regulation of the spatial and temporal patterns of gene expression, including during developmental programs and cardiogenesis [81], postulating the possible intervention of histone modifications in CHD development. Different authors have investigated histone modification as a relevant epigenetic factor in these pathologies [81].

Several functional studies have highlighted the importance of the p300 HAT (p300HAT), which has been implicated in mice cardiac development, and mutations in the HAT domain have been shown to result in the development of VSD [82]. This protein acts as a transcriptional coactivator, regulating gene expression through chromatin loops, while interacting with DNA-binding TFs, including cardiac transcriptional regulators like GATA4, TBX5, and NKX2, among others [83].

Notably, the acetyltransferase activity of p300 has also been shown to alter cell functions and influence cell dysfunction while accelerating cardiac aging. For example, TGF-β-induced fibrogenesis is disrupted in p300-depleted fibroblasts; furthermore, the acetyltransferase domain of p300 is essential for inducing Type I collagen synthesis, even in the presence of similar chromatin-binding proteins. Moreover, TGF-β fails to promote myofibroblast differentiation in the absence of p300, underlining the importance of this protein as a profibrogenic epigenetic regulator, likely through its recruitment to the collagen gene promoter and interaction with the TGF-β-activated Smad2/3/Smad4-Sp1 complex [84], highlighting the importance of p300 in the context of developing heart pathologies.

In this regard, Zhou et al. explored the role of histone acetylation through p300 in GATA4 transcriptional regulation, as well as its effect on heart function and development. They assessed GATA4 expression over time in murine embryos and showed the expression to be regulated by p300 and chromatin-binding protein (CBP)-mediated histone acetylation. The authors verified the role of p300 in cardiogenesis through a knockdown of the protein in cardiac progenitors by RNAi, showing the decrease in GATA4 expression when compared to *TBX5*. Additionally, the authors showed a modulation of H3K4, H3K9, and H3K27 acetylation both in *GATA4* and in the *TBX5* promoter. Finally, the p300 bromodomain, involved in the recognition of acetylated lysine residues, was validated to cause a decrease in GATA4 expression. Therefore, the authors concluded this to be an essential regulatory mechanism in cardiogenesis [83]. Thus, p300 is an essential protein to evaluate in future assays.

Moreover, Leigh et al. identified new epigenetic factors that are drivers of CHD. To this end, they employed a chemical–genetic combined approach. The authors set out from computational modeling, synthesis, and posterior screening and optimization to obtain an inhibitor of the TAF1 bromodomain (TAF1 is a subunit of the transcription initiation complex TFIIID). The effects of bromodomain inhibition on mESCs (mouse embryonic stem cells) and differentiation towards atrial and ventral cardiomyocyte fates were assessed, and it was found to modulate gene expression in cardiomyocyte subtypes, indicating a possible role in cardiogenesis and the activation of gene expression programs in the fetus. The authors also evaluated the effect of genetic modifications on *TAF1* on cardiac gene expression in HEK cells, particularly assessing previously reported genetic variants from ASD and VSD, and in silico analysis predicted that it would have a damaging effect on TAF1 function. Finally, the authors demonstrated TAF1 to have an inhibiting effect on atrial gene expression, which was canceled with the introduction of these genetic variants, suggesting the existence of an epigenetic mechanism that drives the CHDs associated with these variants [85].

Furthermore, other data suggest that TAF1 is an essential factor in embryonic development. TAF1 and other TFIIID components play key roles in cell proliferation and growth. TAF1 is thought to be key in G1-phase cell cycle progression. These roles presumably have cell-state-specific effects, as the expression of these components is higher in muscle cell progenitors, and they seem to regulate stemness [86]. These roles help paint a picture of the role that the epigenetic modulation of TAF1 may have in influencing complex developmental processes like cardiogenesis in the context of CHD.

Despite the relatively little information relating to histone modification in CHDs, small-molecule HDAC inhibitors have been shown to block adverse heart remodeling in animal models with heart insufficiency and hypertrophy. Differential H3K4 and H3K9 trimethylation is associated with reduced Kcnip2 levels, provoking a wearing in the sodium and calcium currents, extending the duration of the action potential, and contributing to heart insufficiency [87]. Furthermore, a recent study corroborated these findings by showing elevated HDAC catalytic activity in univentricular CHD patients, where HDAC inhibitors are suggested as a possible therapy for this kind of CHD [88].

Several concomitant metabolic substrates also promote the expression of the genes responsible for hypertrophy by regulating histone modifications as substrates or enzymatic modifiers, which signals their double function as metabolic and epigenetic regulators. This approach highlights the importance of histone modifications like Acetyl-CoA-dependent acetylation, SIRT-mediated NAD+-dependent acetylation, LSD-mediated FAD+-dependent acetylation, and JMJD-mediated α-KG-dependent acetylation in the context of heart disease [89].

Another epigenetic factor known to be associated with CHD etiology is KAT2B, (lysine acetyltransferase 2B). KAT2B has in vitro and in vivo binding activity with CBP and p300 acetyltransferase, and it is relevant to the TGF-β signaling pathway. Through genetic association studies and sequencing, the genetic variants rs3021408 and rs17006625 on the *KAT2B* gene were described and associated with higher risks of CHD in the Chinese Han population [90]. With this in mind, a study from Gosh et al. reports an interaction between KAT2B and KAT2A with TBX5, which is acetylated by this protein at Lys339, bolstering its transcriptional activity and enabling nuclear retention. The authors also demonstrate a perturbation in zebrafish heart development after *kat2b* and *kat2a* Morpholino-mediated knockdown, which is replicated by CRISPR/Cas-induced mutations in both genes [91]. This provides evidence of the relationship between KAT2B-mediated epigenetic function and heart development.

On the other hand, acetylation has been shown to be an important factor in the context of syndromic CHD. Genitopatellar syndrome and SBBYSS syndrome are caused by a mutation in the KAT6B acetyltransferase gene. Congenital heart defects, cryptorchidism, dental anomalies, and thyroid anomalies are often noted as usual complications of this syndrome [92].

## 7. ATP-Dependent Chromatin Remodeling

ATP-dependent chromatin remodeling complexes perform essential functions in the regulation of gene expression by modifying the organization, structure, and accessibility of chromatin. In the nucleus of eukaryotic cells, the processes of DNA metabolism, including mechanisms of transcription, replication, and DNA repair, occur via DNA packaged into nucleosomes and chromatin structures. Chromatin can be condensed, making it difficult for proteins to access it. For this, cells use enzyme complexes called chromatin remodeling factors (CRFs), which act by catalyzing the ATP-dependent restructuring and repositioning of nucleosomes [93]. These CRFs can modify the position of nucleosomes in regulatory regions on chromatin and are critical for normal gene regulation, because tight interactions between nucleosomes and DNA can prevent the association of DNA with transcription factors and the core transcription machinery [94].

Chromatin remodelers are multi-subunit complexes that share a common SF2 ATPase helicase domain in their catalytic subunit; by using the energy obtained from ATP hydrolysis, they can reposition nucleosomes in chromatin, thus modifying their accessibility and the specific binding to the genome. The catalytic activity of these complexes are mediated through the different associated members of the ATPase complex. There are four main families of chromatin remodeling complexes according to their protein similarity and domain structure: CHD (chromodomain helicase DNA-binding), found in mice; SWI/SNF (switch/sucrose non-fermenting), initially identified in prokaryotes and yeast; ISWI (imitation switch), identified in Drosophila; and INO80 (inositol-requiring 80), discovered in yeast [95,96].

Therefore, the integration of different assessments, such as genetic, biochemical, structural, single-molecule, new omics, and biophysical methods, has provided new insights into epigenetic modulation through chromatin remodeling complexes. Furthermore, it is necessary to emphasize continued precision research in order to discover new drugs to treat diseases involving defects in chromatin remodeling [96]. Given the above, it has been shown that ATP-dependent chromatin remodelers are highly relevant to developmental processes and are involved in various diseases, for instance, CHARGE syndrome.

CHARGE syndrome is a complex neurodevelopmental disorder characterized by different congenital anomalies which form the syndrome’s acronym: ocular coloboma (C), heart malformations (H), atresia of the choanae (A), retardedgrowth (R), genital hypoplasia (G), and ear abnormalities (E). Its clinical phenotype involves a wide spectrum of CHDs, from mild malformations like PDA to more severe phenotypes like TOF, with conotruncal and atrioventricular septal defects being over-represented. Mutations in the ATP-dependent chromatin modifier chromodomain helicase DNA-binding protein 7 gene (*CHD7*) have been identified as a major cause of CHARGE syndrome [96,97], with some studies suggesting an increase in Congenital Heart Disease prevalence in patients with pathogenic *CHD7* variants.

Indeed, pathogenic *CHD7* mutations usually disturb chromatin-modifying activity, implying a possible epigenetic origin for CHARGE syndrome, which could be explained by the role that CHD7 plays in significant developmental pathways such as BMP [94], altering cell migration, and the development of cell lineages like the cardiac mesoderm [97]. Nevertheless, many details about its role in CHARGE syndrome remain to be elucidated, including in the context of CHD and the apparent bias towards some serious phenotypes, and further research is required to understand this topic well. The syndrome mentioned is a specific example of the participation and importance of epigenetic mechanisms in CHD.

## 8. Polycomb and Trithorax Complex Proteins

The multiprotein Polycomb (PcG) and Trithorax (TrxG) complexes act antagonistically to carry out the expression of genes key to the cell differentiation and developmental processes initially described in *Drosophila* [98]; more specifically, PcG-group proteins have a fundamental role as transcriptional repressors, while the members of the TrxG complex group act as transcriptional activators [99]. Figure 1 represents the main epigenetic mechanisms involved in CHD.

The family of repressor PcG proteins is divided into two main complexes: Polycomb repressive complex 1 (PRC1) and Polycomb repressive complex 2 (PRC2). These repressor molecules are crucial for gene transcription regulation through histone modification. The PRC1 enzyme has ligase E3 activity that regulates ubiquitination in the 119 lysine residue of the H2A histone (H2AK119ub), while PRC2 takes care of the mono-, di- and trimethylation of the lysine 27 residue in the histone H3 tail (H3K27), which regulates the equilibrium between self-renewal and ESC differentiation [102,109].

The dynamic binding profile in histone H3 (H3K27me3) mediated by PRC2 has been described as participating predominantly in cardiac cell fate decisions during heart development [110]. Recently, Hanafiah et al. analyzed the structure of loops in the PCR2 through Hi-C by editing mouse embryonic stem cells with CRISPR/Cas9, rendering them in a prepared state of activation–inactivation. Interestingly, the loss of a PCR2 subunit (Polycomb group ring finger protein 2) was found to interrupt the chromatin loops, preventing the activation of essential differentiation genes [111].

The role of Polycomb complexes in heart development is highlighted by studies examining the inactivation of the PRC2 subunit EZH2. Functional models have shown that the loss of EZH2 mediated by Nkx2–5Cre in cardiac progenitors and in cardiomyocytes results in lethal heart abnormalities and disrupts cardiomyocyte gene expression, with the thinning of the compact myocardium and decreased cardiomyocyte proliferation, strongly suggesting that this is a common mechanism by which PRC2 regulates proliferation in development and disease. The mean result of this disrupted mechanism caused lethal congenital heart malformations, namely, compact myocardial hypoplasia, hypertrabeculation, and VSDs [112].

Similarly, studies using a mutant zebrafish model created by introducing a premature stop codon in the rnf2 gene (an essential component of PRC1) displayed defects in the maintenance of cellular identity and organ integrity, proving the impact of epigenetic regulation on cardiac development. This rnf2 mutation impeded its expression in mutant embryos, causing defects in heart development like heart morphological malformations, cardiac looping defects, and problems in ventricle and atrium formation. Furthermore, gene expression analysis of these models revealed the deregulation of key genes involved in organ development, including the heart, showing a relation between rnf2 loss and the alteration in cardiac transcription factor expression, as well as structural and myocardial genes [113].

On the other hand, the TrxG proteins are essential to preserving the transcriptionally active pattern, forming great multiprotein complexes like SWI/SNF (mammalian homolog BAF/PBAF), SWItching defective/sucrose non-fermenting, and COMPASS/COMPASS-like (complex proteins associated with Set1) [101].

The ATP-dependent multiprotein complex SWI/SNF has as a main function of chromatin remodeling through nucleosome displacement in order to create accessible chromatin regions and promote active transcription; for this, complex proteins are recruited to promoter regions.

The COMPASS and COMPASS-like complexes are made up of methyltransferases with a SET domain that catalyze mono-, di-, and trimethylation at lysine 4 in histone 3 (H3K4me1/2/3). The catalytic cores of these complexes are composed of common subunits; every specific complex incorporates a SET domain methyltransferase—Set1 (Set1A or Set 1B), Trx (MLL1 or MLL2), or Trr (MLL3/4)—as well as several additional subunits, which produces distinctive chromatin binding patterns and additional catalytic activities for the complexes [101,103,114].

Regarding the aforementioned complexes, the MLL3 and MLL4 proteins perform a fundamental role in epigenetic regulation and gene activation as they modify histones in enhancer regions, impacting the expression of genes essential to development and cell differentiation. Any alteration in these proteins can have important effects on cell biology and contribute to the appearance of several diseases, including CHDs. Therefore, Gökbuget et al. published a study assessing how the loss of MLL3/4 affects enhancer regulation and gene activation during ESC differentiation [115]. Within this study, the loss of MLL3/4 was shown to result in a dissociation between H3K4 monomethylation (H3K4me1) and H3K27 acetylation (H3K27ac) in the enhancers. Regularly, these marks are coordinated, and both are associated with enhancer activity and gene activation. Nevertheless, H3K4 monomethylation was found to remain in enhancers even in the absence of MLL3/4; however, H3K27 acetylation was considerably reduced. This suggests that MLL3/4 are not strictly necessary for H3K4me1 deposition, but do perform key roles in H3K27ac regulation, impacting essential gene activation [115].

The precise regulation of enhancers is crucial for normal heart development, since these control the activation of essential genes needed for the differentiation of different cardiac cell types as well as their adequate structure. Given that MLL3/4 are fundamental for epigenetic mark coordination in enhancers, their loss was assessed using a MLL3/4 double-knockout (DKO) model to evaluate the impact of MLL3/4 depletion on chromatin structure and transcription during the early differentiation of mouse embryonic stem cells. The analysis revealed that MLL3/4 activity is crucial at most sites where H3K4me1 levels change during differentiation, either by gaining or losing this mark, and their loss might alter the regulation of critical heart development genes, contributing to organ malformations or dysfunctions [116].

Furthermore, Zhu et al. published a study in which they explored the specific functions of the Set1, Trx, and Trr subunits in multiple lines of *Drosophila* flies. The silencing of *Set1*, *Trx*, or *Trr* in the heart resulted in reductions in two types of histone methylation marks on H3K4me1 and H3K4me2, indicating their roles in H3K4 methylation. These findings suggest that Set1, Trx, and Trr play critical roles in regulating histone methylation during heart development. Altogether, each COMPASS complex mediates distinct methylation states and displays preferential times of activity during fly heart development, indicating that these roles may have implications for understanding CHD [117].

## 9. Non-Coding RNAs

In the last few decades, the arrival of modern high-throughput techniques has evidenced the fact that only a small amount of RNA transcribed from DNA is translated into protein. In truth, most transcripts can be classified as non-coding RNAs (ncRNAs) which perform other functions in the cell, including post-transcriptional gene regulation [118]. Diverse types of ncRNAs have been described; one of the most known, mainly in health and disease, is miRNAs, transcripts of between 18 and 25 nucleotides that regulate gene expression by binding to specific transcripts and either directing them to degradation or blocking their translation [104,119]. Each miRNA can have an effect on several genes, which means that even the deregulation of a single one can affect an entire gene network and potentially have pathological effects [120]. Other transcripts have emerged as important actors in gene regulation, including small nucleolar RNAs (snoRNAs), with the main responsibility of mediating RNA maturation occurring via the post-transcriptional modifications of ribosomal RNA (rRNA) and small nucleolar RNAs (snRNAs); therefore, these transcripts have important roles in ribosome and spliceosome function [121].

On the other hand, long non-coding RNAs (lncRNAs) have also become relevant on account of their participation in regulation mediated by different mechanisms, like interactions with other ncRNAs, DNA, mRNA, and proteins [122]. Within this category, circular RNAs (circRNAs) are sometimes mentioned; these are single-stranded RNA molecules with a circular closed loop [123] whose functions include acting as a miRNA sponges, the regulation of gene expression through RNA-binding proteins, or direct interaction with RNA polymerase or protein translation [124]. Here, we address the relation between ncRNAs and CHDs.

### 9.1. MicroRNAs (miRNAs) and CHD

Several reports regarding deregulated miRNAs (Table 2) in CHDs are well known. The pathology for which most of this knowledge has so far been gathered is TOF. In this regard, O’Brien et al. researched ncRNA expression in right ventricle myocardial tissue from 16 TOF pediatric patients. The authors found 61 deregulated miRNAs in the patient group compared with the healthy subjects. In the enrichment analysis, 44 cardiac network genes had significant negative correlations. The miRNAs mainly targeted genes involved in heart development, which are involved in relevant pathways like Wnt, Notch, Sonic Hedgehog Homolog (SHH), and cardiomyocyte differentiation via BMP receptors [106].

On the other hand, Kan et al. explored RNA regulatory networks through the in silico analysis of TOF patients and identified differentially expressed mRNA, circRNA, and miRNA profiles, constructing a regulatory network of 29 miRNAs, 13 circRNAs, and 88 mRNAs. A combination of Gene Ontology and KEGG analysis showed that six hub genes (*IL6R*, *PIK3R1*, *STAT3*, *SOCS3*, *OSMR*, and *BCL2L11*) were significantly enriched in apoptotic-related pathways, including the extrinsic apoptotic signaling pathway, JAK-STAT signaling pathway, and PI3K-Akt signaling pathway. The screened hub genes were crucially involved in hypoxia and apoptosis. In this same study, the authors identified a regulation axis between circRNA has-circ-0060, miRNA hsa-miR-148a, and the *BCL2L11* gene, which is associated with cardiomyocyte apoptosis. In this instance, hsa-miR-148a was observed to be downregulated and BCL2L11 overexpressed, which consequently induced apoptosis [105].

In a different study by Ramachandran et al., the miRNomes of patients with TOF, VSD, and ASD were evaluated through high-throughput deep sequencing. The authors reported 295 deregulated miRNAs, which were functionally associated with signaling pathways involved in proliferation, survival, angiogenesis, migration, and cell cycle regulation. Additionally, through qRT-PCR, they validated the expression of the miRNAs hsa-miR-221-3p, hsa-miR-218-5p, and hsa-miR-873-5p, which are associated with cardiac processes [145].

Moreover, Yang et al. examined the expression profiles of exosomal miRNAs found in the amniotic fluid of women pregnant with TOF-diagnosed fetuses. The authors found 257 differentially expressed miRNAs, of which 25 could regulate Wnt and Notch signaling; moreover, the relevance of these miRNAs has been demonstrated in other studies of TOF. Furthermore, functional studies have shown a significant potential for influencing heart development and contributing to the pathogenesis of CHD. For instance, miR-200a-3p and miR-10a-5p were selected due to their potential role in cardiomyogenic differentiation and transfected in P19 cells. In addition, these cells were induced to differentiate into cardiomyocytes. The results showed that the overexpression of miR-10a-5p negatively regulated the expression of these cardiomyocyte marker genes during differentiation. This finding demonstrates that the upregulation of miR-10a-5p inhibits the expression of key cardiomyocyte markers, particularly *TBX5*, for which a direct miRNA–gene interaction was confirmed [146]. These findings highlight the importance of the cardiogenic genes regulated by these biomolecules in elucidating new molecular mechanisms implicated in CHD.

Additionally, in vitro cardiac differentiation assays have also provided new insights relating to miRNAs in CHD. In this way, Li et al. characterized miRNA expression profiles in differentiated human embryonic stem cells (hESCs) and plasma from ASD patients through RNA sequencing. In this study, the authors reported eight significantly downregulated miRNAs in the differentiated cells, which were overexpressed in plasma from ASD patients. Within these miRNAs, miR-20b-5p was shown to have a role as an inhibition regulator during hESC differentiation by targeting TET2, which is responsible for 5hmC hydroxymethylation and activating transcription factors involved in cardiogenesis. This might represent a new miR-20b-5p/TET2 regulatory mechanism during heart differentiation and potentially a new ASD marker [147].

A study conducted by Sucharov et al. assessed miRNA expression in the right ventricle myocardium from HLHS pediatric patients. In this population, the authors described a profile of 93 significantly deregulated miRNAs. Furthermore, they validated the expression of some of these miRNAs (miR-100, miR-145a, miR-99a, miR-137-3p, miR-204) by qPCR. Interestingly, the expression of these miRNAs correlated with the stage of surgical treatment, suggesting that the expression of these miRNAs could be regulated by the right ventricle volume charge. Lastly, through in silico analysis, they identified nine miRNAs that were downregulated and four miRNAs that were upregulated in HLHS; the pathway analysis highlighted the PI3K-AKT, MAPK, and WNT signaling pathways. In addition, *QKI*, *CDK6*, *BAZ2A*, *FOG-2*, and *SOX11* were identified as potential cardiovascular targets. Therefore, the authors suggested a relation between the miRNA profile and the development of this pathology [128].

### 9.2. CHD and Other ncRNAs

Some studies have addressed the relation between snoRNAs and CHDs. In this regard, O’Brien et al. identified 135 differentially expressed snoRNAs in myocardial tissue from patients with ToF. Most of the studied snoRNAs were downregulated and had the spliceosomal snRNA U6 as a target, but not U12. Therefore, the authors investigated the expression differences between alternative splicing variants in tissues with TOF and healthy tissues in between the exon/intron U2 and U12 recognition sections, and they found a much more pronounced effect in slicing variants from U2-type exons than in U12 when assessing critical heart development genes. Furthermore, the U2 and U12 snRNAs were downregulated in ToF myocardial tissue. These findings suggest a relation between the low expression of snoRNAs that target the U6 and U2 snRNAs, alternative splicing, and TOF patients [106].

Additionally, studies relating to lncRNAs with CHDs have recently been performed. In this regard, Ma et al. analyzed the Gene Expression Omnibus (GEO) and performed in silico analysis to characterize lncRNA and mRNA expression in fetal and adult hearts, identifying regulator regions involved in CHD. The authors then selected *lncRNA SAP30-2:1*, as it was upregulated in fetal tissue and because it possibly acts upon the *HAND2* gene, given their proximity. The expression analysis of this lncRNA in tissues from patients with TOF, VSD, or double-chambered right ventricles revealed a decrease in expression compared with healthy tissues. Afterwards, the authors performed in vitro assays and reported the knockdown of *SAP30-2:1* to inhibit cell proliferation and cause a decrease in *HAND2* expression, possibly through physical interaction, which exposed a mechanism by which *SAP30-2:1* impacts *HAND2* expression, a key gene in cardiogenesis [107].

Moreover, Zhang et al. obtained lncRNA and mRNA expression profiles from 22 cardiac tissue samples from patients with TOF by transcriptomic analysis. Subsequently, through a causal interference framework based on expression correlations and validated evidence of miRNAs-lncRNA-mRNA, they created the lncRNA-driven competing endogenous RNA (ceRNA)-mediated network, in which they identified four core lncRNAs (FGD5-AS1, lnc-GNB4-1, lnc-PDK3-1, and lnc-SAMD5-1). After validating FGD5-AS1 for its relevance to CHD, it was shown to regulate SMAD4. Both FGD5-AS1 and SMAD4 were able to bind with hsa-miR-421, as investigated through dual-luciferase reporter gene assays. Interestingly, FGD5-AS1 inhibition influenced a reduction in SMAD4 expression in two cell lines [HEK 293 and fetal heart cell line (CCC-HEH-2)]. Furthermore, an increase in hsa-miR-421 transcription was observed in both cells. This demonstrates the importance of lncRNA regulation [137].

Recently, Lu et al. studied lncRNAs encoded within copy number variants (CNVs) associated with CHDs. For this purpose, they built coexpression networks between CNV-lncRNAs and protein-coding genes. Through a complex analysis, the authors identified *HSALNG0104472* as a hub CNV-lncRNA in a network module enriched with non-syndromic CHD genes. Afterwards, in vitro assays were performed, through which *HSALNG0104472* was observed to act in the regulation of cardiac differentiation, offering a better understanding of cardiac defects related to the deletion of the 15q11.2 region, within which this lncRNA is located, and highlighting the relevance of non-coding regions in the possible pathogenesis of CHD-associated CNVs [108].

On the other hand, circRNAs have become a main interest for some researchers. In this regard, Wu et al. described three circARNs that were downregulated in pediatric CHD patients—has-circRNA-004183, has-circRNA-079265, and hsa-circRNA-105039—which were identified as possible biomarkers through ROC curve analysis. Furthermore, the authors developed a coexpression network with a total of 43 circRNAs, 9 mRNAs, and 29 miRNAs. The network revealed a strong correlation, and miR-17-5p was predicted to target *WNT5A*, *MEF2C*, *TBX3*, and *HOXA3*; also, miR-20b-5p was found to potentially target both *TBX3* and *HOXA3*, while miR-193-3p specifically targeted *HOXA3*, and miR-24-3p targeted *WNT5A*, highlighting regulating processes like apoptosis, differentiation, and cardiogenesis [148].

## 10. Limitations

The present review shows recent contributions, through an epigenetic approach, to CHD research, showing progress within this field as well as new challenges for scientific research in it. Nevertheless, some of these studies share methodological challenges. For instance, (A) the reduced sample size of the evaluated groups, which may in part be due to the complexity of obtaining biological material, particularly in the pediatric population control group [69,71].

(B) Study design. Several of the examined studies are transversal and/or comparative. Therefore, the absence of a longitudinal follow-up has, so far, not allowed researchers to establish causality in this type of pathology, which makes it a challenge to settle whether the observed epigenetic alterations are causal events or consequences of the heart defect, as well as evaluating their persistence or variability through time [74,75,128]. This kind of approach has been an understandable methodological restriction, given that longitudinal studies in pediatric populations carry with them several challenges, especially when they involve interventions through repeated molecular analyses.

(C) Functional studies. Some studies describe important associations between epigenetic signatures and CHDs; however, they do not allow us to precisely know the impact of the reported association, which means that functional experimentation is required to elucidate the mechanism or biological pathway which could be involved in the given pathology [74,149].

(D) Experimental in vitro and animal models have been fundamental to elucidating molecular mechanisms under controlled laboratory settings; nevertheless, their extrapolation to a human context requires additional validations, especially given the limited availability of in vivo human heart tissue. This restriction answers to ethical and logistical considerations, which are particularly critical when dealing with samples from early stages of embryonic development [86,89].

To summarize, these considerations signal the areas in which to improve and suggest key avenues to consolidate molecular-level cardiology in CHD in order to design future studies with greater populations, longitudinal approaches, and with the use of multi-omic technologies. Adding this analytical perspective strengthens the interpretation of the available findings and allows for the clear projection of research guidelines, which may lead to clinically significant findings that can be applied to translational medicine.

## 11. Perspectives

As previously mentioned, CHDs are a group of complex and multifactorial pathologies and epigenetic mechanisms that represent an area of opportunity to continue generating scientific research in a field that is still underexplored but of high relevance. Therefore, knowledge of the methylation state in different regions of the genome is essential, as it will allow for the precise establishment of clinically relevant implications, prevention strategies, timely diagnosis, and personalized treatments that can help us build towards translational medicine that positively impacts patients’ quality of life and prognosis.

Within the useful methods of epigenetic analyses of complex illnesses is the use of epigenetic clocks, high-sensitivity markers of biological aging, that can be used in several tissues through certain methylation sites by comparing chronological age to biological age. Therefore, this tool may allow us to assess accelerated aging in CHD. Epigenetic clocks offer a promising avenue to evaluate prognosis as well as to predict disease severity and progression, and specific epigenetic clocks could be established for different populations [150,151]. Hence, epigenetic clocks are useful tools to identify people at risk of aging-related illnesses [152].

Non-coding RNAs have also been recognized for their potential as biomarkers in disease. miRNAs, in particular, have the advantage of being easily accessible, as they can be extracted from bodily fluids like blood and they usually have a high specificity for their tissue and cell-type provenance, as well as varying levels in different states of health and disease. While this application of miRNAs is still in its early stages of development, research is ongoing, and techniques for miRNAs analysis and identification are relatively accessible, as they consist of widely used methods like RT-qPCR, Northern blot, or in situ hybridization [153]. In CHD, miRNAs may have a high potential as biomarkers for diagnosis and personalized medicine strategies. As previously shown, this could even be accomplished via exosomal-derived miRNAs from amniotic fluid [146].

Despite these advancements, the challenges for future research include the study of the temporal and spatial control of these epigenetic factors, particularly in processes of cardiac regeneration and repair. An innovative strategy involves selectively suppressing epigenetic modifiers during key stages of cardiac development to explore new therapeutic possibilities. This suggests that combining epigenetic drugs with conventional treatments could optimize clinical outcomes and reduce long-term complications in patients with CHD [154]. Several approaches are being developed in the field of epigenetics, including the inhibition of epigenetic modulators, dCas9 fusions with transcriptional regulators, and the use of non-coding RNAs like siRNAs or miRNAs [155]. As this field continues to grow, these approaches may appear as a potential therapeutic tools for treating the development of CHDs in the future.

On the other hand, it is important to note that the presented epigenetic mechanisms exist within a complex network where they can interact, promoting or inhibiting their effects [156]. Therefore, it is essential to understand and characterize the crosstalk between these epigenetic marks in order to fully understand their effects on pathologies such as CHD. DNA methylation patterns and histone modifications are strongly correlated [156]. DNMT enzymes have been found to interact in regulatory loops with histone methyltransferases in a bidirectional and mutually reinforcing manner [157]. For instance, evidence suggests that DNA methylation could reinforce gene silencing initiated by histone modifications, as DNMT3A and DNMT3B were found to primarily methylate nucleosomal DNA, directed by the catalytically inactive DNMT3L, apparently in the absence of H3K4 methylation [158]. Similarly, the H3K4me3 mark has been shown to bind to the DNMT3A/DNMT3B domain, inhibiting enzymatic activity [159]. A few ncRNA species are also known to interact with chromatin; these are known as chromosome-associated regulatory RNAs (carRNAs), and they contribute to genome organization as well as transcript regulation [159]. The promoter-associated lncRNA upper hand (Uph) is an example of particular relevance to CHDs, as it provides a non-coding transcriptional regulatory mechanism for Hand2 expression in cardiac tissue, probably through the maintenance of the super-enhancer signature, including histone acetylation, which controls Hand2 expression [160]. These examples highlight how future research into epigenetic mechanisms involved in diseases, including CHD, should include the study of epigenetic crosstalk, which can be further enhanced by the use of genomic and proteomic tools such as ChIP-seq, ATAC-seq, or Mass Spectrometry [156].

## 12. Conclusions

Currently, mechanisms of epigenetic regulation represent a new avenue of knowledge in the clinical and molecular aspect of CHDs, where cardiogenesis is a central axis, as they involve specialized epigenetic regulation processes that allow for the early expression of critical genes needed for the correct orchestration of the developmental pathways and correct cardiac working.

Therefore, research on the epigenetics of these pathologies is still necessary, as they remain relatively unexplored despite their high global frequency and the fact that around 90% of patients reach an adult age. Future research employing experimental cell models will be indispensable tools in biomedical research, as they provide valuable information about cell mechanisms, disease pathogenesis, and the application of therapeutic development.

Also, the implementation of single-cell epigenomics could allow us to know the epigenetic signature of individual cells, which could drive us towards a more precise comprehension of epigenetic marks, cell heterogeneity, and gene regulation in cells and diseases, including in CHD. Furthermore, this new knowledge may be used with a direct focus on epigenetic editing, meaning the manipulation of epigenetic marks in specific genomic loci with the purpose of investigating specific gene functions by silencing or activating them in concrete locations. Emerging epigenetic knowledge can be greatly useful to form the basis of, develop, and approach new treatments and therapeutic strategies based on epigenetic manipulation in order to benefit patient attention.

## Figures and Tables

**Figure 1 cells-14-00820-f001:**
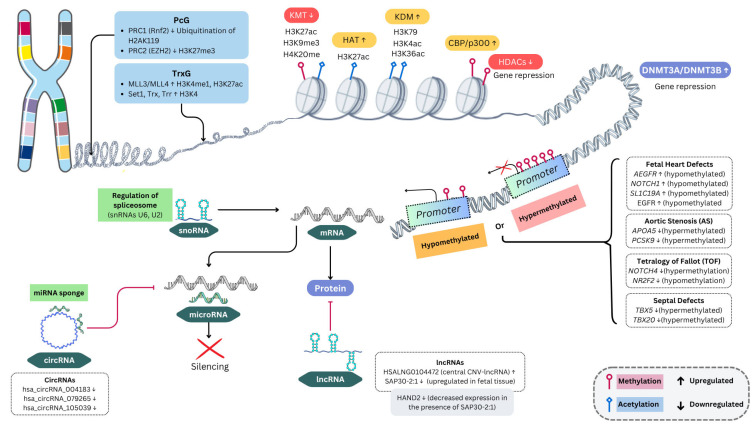
Epigenetic mechanisms in CHDs. Representation of main epigenetic mechanisms involved in CHD. Polycomb (PcG) and Trithorax (TrxG) complexes modulate chromatin accessibility by repressing or activating gene expression through post-translational histone modifications. H3K27me3 and H2AK119 ubiquitination repress transcription, while H3K4me1, H3K27ac, and H3K4ac promote gene activation. DNA methylation, mediated by DNMT3A/DNMT3B, silences gene expression, while histone acetylation (HATs, CBP/p300) eases transcriptional activation. Additionally, non-coding RNAs (snoRNAs, microRNAs, lncRNAs, and circRNAs) regulate mRNA stability and gene expression, impacting heart development. Alterations of methylation patterns have been associated with different types of CHD, like TOF, AS, and septal defects [9,74,78,83,89,100,101,102,103,104,105,106,107,108].

**Table 1 cells-14-00820-t001:** The environment and Congenital Heart Diseases.

Gestational Environmental Exposure	Relevance	Associated Pathology	Reference
Polycyclic aromatic hydrocarbons (PAHs)	Exposure to PAHs can interfere with proper cardiovascular development in fetuses. The combination of chemical compounds in tobacco fumes, vehicle exhausts, and certain industries’ emissions can induce oxidative stress and inflammation in fetal tissues. Since PAHs are lipophilic, they can cross through cell membranes as well as the placenta. In the fetus, they form intermediate reagents that bind covalently to DNA.	Conotruncal defects, obstruction of right ventricle outflow tract, ASD, TOF	[25]
Air pollutants (CO_2_, SO_2_)	Exposure to air pollutants can induce an inflammatory response and increase oxidative stress, which might alter placental circulation, affect fetal oxygenation, and disturb normal heart development. Interference with the development of fetal blood vessels affects heart perfusion, altering gene expression in heart cells.	ASD, CoA, TOF, PDA, PS	[26]
Pesticides	Pesticides are associated with fetal heart development by acting as endocrine disruptors, generating oxidative stress, altering gene expression (*VEGF* and *NOTCH*), and inducing maternal inflammation. They can interfere with cell migration from the neural crest, favoring conotruncal defects. They can also alter the equilibrium between the proliferation and apoptosis of key cardiac cells, generating alterations in angiogenesis and gene expression.	TOF, HLH, PS, VSD, ASD	[27,28]
Drugs			
Valproic acid	Valproic acid causes an alteration in the expression of genes associated with cell polarity (*Vangl2*, *Scrib*) and dysfunction in the activities of histone deacetylases (HDAC1/2/3), which might interfere with the proper formation and closure of the interventricular septum.	VSD	[29]
Fluoxetine	Exposure to fluoxetine may induce an alteration of serotonin regulation, which affects heart and blood vessel development in fetuses. This can interfere with placental circulation, alter gene expression in heart cells, and disturb normal heart development.	Subaortic stenosis, secundum ASD, muscular VSD, CoA	[30]
Citalopram	Citalopram can affect serotonin regulation, altering fetal blood vessel development. This might interfere with adequate fetal heart perfusion, altering normal development and increasing the risk of septal defects.	Subaortic stenosis, secundum ASD, muscular VSD, CoA	[30]
Venlafaxine	Venlafaxine could induce changes in the regulation of fetal blood vessels, affecting placental circulation and causing alterations in fetal cardiac perfusion. This could interfere with heart development and increase the risk of cardiovascular defects like ductal constriction.	Subaortic stenosis, secundum ASD, muscular VSD, CoA	[30]
Escitalopram	Like citalopram, escitalopram can interfere with serotonin regulation. This can alter blood vessel development and placental circulation, increasing the risk of heart defects.	Subaortic stenosis, secundum ASD, muscular VSD, CoA	[30]
Albuterol	Bronchodilators, like albuterol, are drugs that act on the beta-adrenergic receptors in the lungs to dilate the respiratory tract. Excessive or early bronchodilator use may alter this signaling processes and result in defects of key heart structures like valves or septums.	Truncus Arteriosus, interauricular secondary communication	[31]
Lithium	Lithium can affect cellular calcium homeostasis, which is crucial for the development of cardiac and vascular structures. It has been suggested that it can interfere with G-protein-dependent signaling. Furthermore, lithium inhibits inositol monophosphatase and inositol polyphosphate 1 phosphatase. This cycle is crucial in cell signaling and growth and development regulation, as well as interfering with Wnt/beta-catenin signaling.	Obstruction of right ventricle outflow tract (RVOTO) and EA	[32]
Others			
Viral infection	Rubella and cytomegalovirus are known human teratogens that can cause birth defects, including cardiac malformations. These infections induce a maternal inflammatory response that affects development, altering the formation of the heart and other organs. Moreover, they can interfere with the formation of fetal blood vessels or affect placental circulation, which contribute to fetal cardiac insufficiency. The drugs used to treat these infections, like antibiotics and analgesics–antipyretics, can also have teratogenic effects.	Conotruncal defects, PDA, peripheral PS	[33,34,35]
Alcohol	Alcohol has been associated with histone hypoacetylation, affecting the expression of development-related genes. Furthermore, it alters retinoic acid biosynthesis and signaling, as well as Wnt and BMP signaling.	TOF, VSD, atrioventricular channel malformation, dextro-transposition of great arteries (RTGA)	[36]

**Table 2 cells-14-00820-t002:** Congenital Heart Diseases and associated miRNAs.

Pathology	Involved miRNAs	Target Genes	References
TOF	miR-27b, miR-421, miR-1275, miR-122, miR-1201, miR-22, miR-222, miR-375, miR-138, miR-421, miR-1, miR-206, miR-940, hsa-miR-148a, hsa-miR-221-3p, hsa-miR-218-5p, hsa-miR-873-5p, miR-19, let-7e-5p, miR-10a-5p, miR-181c, miR-940, miR-181, miR-130, miR-146b-5p, miR-29c, miR-720, miR-424, miR-660, miR-708, miR-363, miR-337-5p, miR-155, miR-154	*SOX4*, *BCL2L11*, *TBX5*, *CDK9*, *FN1*, *MAPK1*	[11,119,125,126,127]
HLHS	miR-30, miR-100, miR-378, miR-99a, miR-145a, miR-208, miR-204	*QKI*, *FOG2*, *CDK6*, *SOX11*, *BAZ2A*	[11,128]
ASD	miR-20b-5p, hsa-miR-19b, hsa-miR-375, hsa-miR-29c, miR-139-5p, miR-196-a2, miR-9, miR-30a, hsa-let-7a, hsa-let-7b, hsa-miR-486, miR-29, miR-143/145, miR-17-92, miR-106b-25, miR-503/424	*ACTC1*, *TBX5*, *PTEN*, *VEGFR-1*	[129,130,131,132]
VSD	miR-1/2, miR-1/1, miR-181c, miR-92, let-7e-5p, miR-155-5p, miR-222-3p, miR-379-5p, miR-409-3p, miR-433, miR-487b, miR-498	*GJA1*, *SOX9*, *BMPR2*,	[11,120]
BAV	miR-26a, miR-95, miR-30b, miR-141	*BMP2*, *ALPL*, *SMAD1*, *SMAD3*	[120,133,134]
TGA	has-let-7e, miR-16, miR-25, miR-18a, miR-93, miR-106a, miR-451, miR-486-3p, miR-486-5p	*ATM*, *PTEN*, *BCL11A*, *FOXO1*, *MMP19*, *IGF1*, *HAT1*, *SMAD1*	[11,135]
Down Syndrome	miR-99a, has-let-7c, miR-125b2, miR-155, miR-802	*IL10*, *NOX4*, *RUNX3*, *CYP24A1*	[120,125,136,137,138,139,140]
DiGeorge Syndrome	miR-23, miR-363, let-7g, miR-361-5p, miR-324-5p, miR-194, miR-720, miR-150, miR-15b-3p, miR-185	*SOX17*, *AFP*, *G3BP2*	[141,142,143,144]

## Data Availability

No new data were created or analyzed in this study.

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
