# Peer review of "Congenital Heart Diseases: Recent Insights into Epigenetic Mechanisms"

_cells, 2025, doi:10.3390/cells14110820_

Round 1
Reviewer 1 Report
Comments and Suggestions for Authors
This manuscript provides a comprehensive overview of the emerging role of epigenetic mechanisms in the pathogenesis of Congenital Heart Diseases (CHD). The topic is timely and significant. The review is generally well-structured and informative, particularly with the inclusion of helpful tables and a figure. However, several areas could be strengthened to enhance its impact and depth.
- The manuscript predominantly summarizes findings. A more critical appraisal differentiating correlative data (e.g., association studies) from evidence supporting causality (e.g., functional studies in relevant models) is needed.
- The review treats different epigenetic layers (DNA methylation, histone modifications, ncRNAs) largely in isolation. Incorporating a discussion on the interplay and crosstalk between these mechanisms would offer a more sophisticated and biologically accurate perspective on epigenetic regulation in CHD.
- The manuscript focuses heavily on fundamental mechanisms. Expanding on the clinical relevance, translational potential (e.g., biomarkers for diagnosis/prognosis, therapeutic avenues targeting epigenetic pathways), and associated challenges would significantly increase the manuscript's scope and impact.
- Where evidence allows, provide more detailed molecular pathways linking specific epigenetic changes to altered cellular behaviors (proliferation, differentiation, migration) and subsequent structural heart defects, moving beyond simply stating "altered gene expression."
- Briefly acknowledge limitations of key cited studies (e.g., design, sample size) or conflicting findings in the literature, where applicable, to provide a more balanced perspective.
- The conclusion regarding future research could be more concrete. Highlight specific key knowledge gaps, promising research avenues, or necessary technological advancements rather than general statements.
Author Response
REVIEWER 1
We thank to the reviewer for the insightful comments and helpful criticisms. We have responded to all comments as detailed below and we now hope that you will find our revised manuscript acceptable for publication. The answers included in the manuscript are marked in blue.
Comments and Suggestions for Authors
This manuscript provides a comprehensive overview of the emerging role of epigenetic mechanisms in the pathogenesis of Congenital Heart Diseases (CHD). The topic is timely and significant. The review is generally well-structured and informative, particularly with the inclusion of helpful tables and a figure. However, several areas could be strengthened to enhance its impact and depth.
- The manuscript predominantly summarizes findings. A more critical appraisal differentiating correlative data (e.g., association studies) from evidence supporting causality (e.g., functional studies in relevant models) is needed.
Answer:
We agree with the reviewer on this important recommendation. In this regard, we have expanded on the requests made in the studies that conducted functional studies on the different epigenetic mechanisms described. Given this, the response is comprehensive, and to avoid confusion, we have described the lines by section where the responses are found in the revised version of the manuscript and properly referenced. Thank you again.
- DNA Methylation
LINES 292 – 305
LINES 307 – 317
LINES 326 – 336
LINES 358 – 366
- Histone modification
LINES 439 – 450
LINES 466 – 472
LINES 488 – 504
- Polycomb and Trithorax complex proteins
LINES 584 – 607
LINES 637 – 654
- Non-coding RNAs
LINES 681 – 686
LINES 695 – 706
LINES 714 – 727
LINES 738 – 749
LINES 774 – 785
LINES 795 – 804
- The review treats different epigenetic layers (DNA methylation, histone modifications, ncRNAs) largely in isolation. Incorporating a discussion on the interplay and crosstalk between these mechanisms would offer a more sophisticated and biologically accurate perspective on epigenetic regulation in CHD.
Answer: We agree with this important recommendation from the reviewer. In order to clarify this issue, we added information in the perspectives section and new references were added.
Please check LINES 876 – 897.
On the other hand, it is important to note that the presented epigenetic mechanisms exist within a complex network where they can interact, promoting or inhibiting their effects [157]. Therefore, it is essential to understand and characterize the crosstalk be-tween these epigenetic marks to fully understand their effect on pathologies such as CHD. DNA methylation patterns and histone modifications are strongly correlated [157]. DNMT enzymes have been found to interact in regulatory loops with histone methyl-transferases in a bidirectional and mutually reinforcing manner [158]. For instance, evidence suggests that DNA methylation could reinforce gene silencing initiated by histone modifications, as DNMT3A and DNMT3B were found to primarily methylate nucleosomal DNA, directed by the catalytically inactive DNMT3L, apparently in the absence of H3K4 methylation [159]. Similarly, the H3K4me3 mark has been shown to bind to the DNMT3A/DNMT3B domain, inhibiting enzymatic activity [160]. A few ncRNA species are also known to interact with chromatin; these are known as chromosome-associated regulatory RNAs (carRNAs), and they contribute to genome organization as well as transcript regulation [160]. The promoter-associated lncRNA upper hand (Uph) is an example of particular relevance to CHDs, as it provides a non-coding transcriptional regulatory mechanism for Hand2 expression in cardiac tissue, probably through the maintenance of the super-enhancer signature, including histone acetylation, which controls Hand2 expression [161]. These examples highlight how future research into epigenetic mechanisms involved in diseases including CHD should include the study of epigenetic crosstalk, which can be further enhanced by the use of genomic and proteomic tools such as ChIP-seq, ATAC-seq, or Mass Spectrometry [157].
- The manuscript focuses heavily on fundamental mechanisms. Expanding on the clinical relevance, translational potential (e.g., biomarkers for diagnosis/prognosis, therapeutic avenues targeting epigenetic pathways), and associated challenges would significantly increase the manuscript's scope and impact.
Answer: We are in line with this important observation. Now, we included information regarding the reviewer's suggestion in the perspectives section.
Please check LINES 838 – 875.
- Perspectives
As previously mentioned, CHD are a group of complex and multifactorial pathologies and epigenetic mechanisms represent an area of opportunity to continue generating scientific research in a field that is still underexplored but of high relevance. Therefore, knowledge of the methylation state in different regions of the genome is essential, as it will allow a precise establishment of clinically relevant implications, prevention strategies, timely diagnosis and personalized treatments that can help us build towards translational medicine that positively impacts the patients’ life quality and prognosis.
Within the useful epigenetic analyses in complex illnesses is the use of epigenetic clocks, high sensitivity markers of biological aging, that can be used in several tissues through certain methylation sites, by comparing chronological age to biological age. Therefore, this tool may allow us to assess accelerated aging in CHD. Epigenetic clocks offer a promising avenue to evaluate prognosis as well as predicting disease severity and progression and specific epigenetic clocks could be established for different populations. [151,152]. Hence, epigenetic clocks are a useful tool to identify people at risk of aging related illnesses [153].
Non-coding RNAs have also been recognized for their potential as biomarkers in disease. MicroRNAs (miRNAs), in particular, have the advantage of being easily accessible, as they can be extracted from bodily fluids like blood and they usually have a high specificity for their tissue and provenance of cell types, as well as varying levels in different states of health and disease. While this application of miRNAs is still on its early stages of development, research is ongoing and techniques for miRNA analysis and identification are relatively accessible, as they consist of widely used methods like RT-qPCR, Northern blot or in-situ hybridization [154]. In CHD, miRNAs may have a high potential as biomarkers for diagnosis and personalized medicine strategies. As previously shown, this could even be accomplished via using exosomal-derived miRNAs from amniotic fluid [146].
Despite the advancements, the challenges for future research include the study of the temporal and spatial control of these epigenetic factors, particularly in processes of cardiac regeneration and repair. An innovative strategy involves selectively suppressing epigenetic modifiers during key stages of cardiac development, to explore new therapeutic possibilities. This suggests that combining epigenetic drugs with conventional treatments could optimize clinical outcomes and reduce long-term complications in patients with CHD [155]. Several approaches are being developed in the field of epigenetic, including inhibitors of epigenetic modulators, dCas9 fusions with transcriptional regulators or the use of non-coding RNAs like siRNAs or miRNAs [156]. As this field continues to grow, these approaches may appear as a potential therapeutic tool for treating the development of CHD in the future.
- Where evidence allows, provide more detailed molecular pathways linking specific epigenetic changes to altered cellular behaviors (proliferation, differentiation, migration) and subsequent structural heart defects, moving beyond simply stating "altered gene expression."
Answer: Thanks to the reviewer for this suggestion to improve the manuscript. Now, we provided additional molecular pathways required.
Please check LINES 318 - 325
Notch signaling is known to be involved in several cellular processes like proliferation, cell death and cell fate decisions. Thus, this pathway is involved in the development of most organs, including cardiogenesis. During heart development, notch signaling is continuously functioning across the processes of cardiac specification, endocardium patterning, valve and chamber morphogenesis and even cardiac regeneration, often also interacting with other relevant developmental pathways like WNT and BMP. Therefore, it stands to reason that differential expression of NOTCH genes like NOTCH4 could be a decisive factor in the development of CHD [70].
Please check LINES 430 - 438
Notably, the acetyltransferase activity of p300 has also been shown to alter cell functions and influence cell dysfunction while accelerating cardiac aging. For example, TGF-β-induced fibrogenesis is disrupted in p300-depleted fibroblasts; furthermore, the acetyltransferase domain of p300 is essential for inducing Type I collagen synthesis, even in the presence of similar chromatin-binding proteins. Moreover, TGF-β fails to promote myofibroblast differentiation in the absence of p300, underlining the importance of this protein as a profibrogenic epigenetic regulator, likely through its recruitment to the collagen gene promoter and interaction with the TGF-β-activated Smad2/3/Smad4-Sp1 complex [84], highlighting the importance of p300 in the context of developing heart pathologies.
Please check LINES 466 - 472
Furthermore, other data suggest that TAF1 is an essential factor in embryonic development. TAF1 and other TFIIID components play key roles in cell proliferation and growth. TAF1 is thought to be key in G1 phase cell cycle progression. These roles presumably have cell state-specific effects, as the expression of these components is higher in muscle cell progenitors, and they seem to regulate stemness [87]. These roles help paint a picture of the role that epigenetic modulation of TAF1 may have in influencing complex developmental processes like cardiogenesis in the context of CHD.
- Briefly acknowledge limitations of key cited studies (e.g., design, sample size) or conflicting findings in the literature, where applicable, to provide a more balanced perspective.
Answer: We agree with the reviewer for this suggestion. Now, we included information regarding this point in the Limitations section.
Please check LINES 807 - 837
- Limitations
The present review shows recent and found contributions, through an epigenetic approach, on CHD; which shows progress within this field as well as new challenges for scientific research in it. Nevertheless, some of these studies share methodological challenges. For instance, (A) A reduced sample size on the evaluated groups, which may in part be due to the complexity in obtaining biological material; particularly in the pediatric population control group [69,71].
(B) Study design. Several of the studies performed are transversal and/or comparative. Therefore, the absence of a longitudinal follow-up has, so far, not allowed to establish causality in this type of pathology, which makes it a challenge to settle whether the observed epigenetic alterations are causal events or consequences of the heart defect, as well as evaluating their persistence or variability through time [74,75,128]. This kind of approach has been an understandable methodological restriction, given that longitudinal studies in pediatric populations carry with them several challenges, especially when they involve interventions through repeated molecular analyses.
(C) Functional studies. Some studies describe important associations between epigenetic signatures and CHDs; however, they don’t allow to precisely know the impact of the reported association, which means functional experimentation is required to elucidate the mechanism or biological pathway which could be involved in the pathology [74,150].
(D) Experimental in vitro and animal models have been fundamental to elucidate molecular mechanisms under controlled laboratory settings; nevertheless, their extrapolation to a human context requires additional validations, especially given the limited availability of in vivo human heart tissue. This restriction answers to ethical and logistical considerations, particularly critical when dealing with samples from early stages of embryonic development [86,89].
To summarize, these considerations signal the areas to improve and suggest key avenues to consolidate molecular-level cardiology in CHD to design future studies with greater populations, longitudinal approaches and with the use of multi-omic technologies. Adding this analytical perspective strengthens the interpretation of the available findings and allows a clear projection of research guidelines which may lead to clinically significant findings to apply in translational medicine.
- The conclusion regarding future research could be more concrete. Highlight specific key knowledge gaps, promising research avenues, or necessary technological advancements rather than general statements.
Answer:
Please check LINES 898 - 918
- Conclusions
Currently, mechanisms of epigenetic regulation represent a new avenue of knowledge in the clinical and molecular aspect of CHD, where cardiogenesis is a central axis, as it involves specialized epigenetic regulation processes that allow early expression of critical genes needed for a correct orchestration of the developmental pathways and correct cardiac working.
Therefore, research on the epigenetics of these pathologies is still necessary, as they remain relatively unexplored despite their high global frequency and the fact that around 90% of patients reach an adult age. Future research employing experimental cell models will be indispensable tools in biomedical research, as they provide valuable information about cell mechanisms, disease pathogenesis and the application of therapeutic development.
Also, the implementation of single-cell epigenomics could allow us to know the epigenetic signature of individual cells, which could drive us towards a more precise comprehension of epigenetic marks, cell heterogeneity and gene regulation in cell and disease, including CHD. Furthermore, this new knowledge may be used towards a direct epigenetic edition focus; meaning, manipulation of the epigenetic marks in specific genomic loci with the purpose of investigating specific gene function by silencing or activating them in concrete locations. Emerging epigenetic knowledge can be greatly useful to base develop and approach new treatments and therapeutic strategies based on epigenetic manipulation to benefit patient attention.

Reviewer 2 Report
Comments and Suggestions for Authors
In the review, the authors described aberrant epigenetic mechanisms in connection with pathogenesis of congenital heart diseases. The manuscript is relevant and appropriate to the literature presented, including identifying gaps in knowledge, interpreting findings and the significance of other recent research on the topic, and providing a readable conclusion. Although the findings are impressive, I would like to make comments regarding their interpretations.
- A short description of the benefits of 3rd generation sequentces & GWAS technology is needed to be added to the section Introduction.
- The authors could add to the text of the paper a discussion about CHARGE syndrome in the context of epgenetic regulation.
- ATP-dependent chromatin modification is required to be described abd discussed in the subsection 2.
Author Response
REVIEWER 2
We thank to the reviewer for the insightful comments and helpful criticisms. We have responded to all comments as detailed below and we now hope that you will find our revised manuscript acceptable for publication. The answers included in the manuscript are marked in blue.
Comments and Suggestions for Authors
In the review, the authors described aberrant epigenetic mechanisms in connection with pathogenesis of congenital heart diseases. The manuscript is relevant and appropriate to the literature presented, including identifying gaps in knowledge, interpreting findings and the significance of other recent research on the topic, and providing a readable conclusion. Although the findings are impressive, I would like to make comments regarding their interpretations.
- A short description of the benefits of 3rd generation sequencing & GWAS technology is needed to be added to the section Introduction.
Answer: Thanks to the reviewer for the suggestion. Now, the following information was added in the Introduction section.
Please check LINES 85-99.
Currently, the use of genomic technologies that allow the identification of a wide range of genetic defects, like Third Generation Sequencing (TGS) and Genome Wide Association Studies (GWAS) have allowed significant progress in the field of genomic medicine. Particularly, TGS involves wide and high quality reads, permitting a better comprehension of genomic structure and avoiding previous amplification by PCR. This technique is already applied in several studies involving genetic diseases and recently, they have been applied to genome assembly, structural variant detection, epigenetic analysis, and pharmacogenomics (PGx) profiling studies [12,13].
Furthermore, GWAS represent a fundamental tool for identifying genetic variants associated with complex diseases and other relevant phenotypes. These studies include wide arrays of individuals. At the moment, the use of GWAS has positively impacted research into multifactorial illnesses, identifying new robust associations of several alleles influencing disease phenotypes in a highly reproducible manner [14,15]. The arrival of these tools, has therefore enhanced research into the genetics and epigenetics of CHDs, making them essential for advancing research in this field.
- The authors could add to the text of the paper a discussion about CHARGE syndrome in the context of epigenetic regulation.
Answer: We agree with the reviewer's recommendation is relevant and necessary to improve the quality of the review. Below, we added the information required in section 7.
Please check LINES 534- 552.
CHARGE syndrome is a complex neurodevelopmental disorder characterized by different congenital anomalies, which form the syndrome acronym: ocular coloboma (C), heart malformations (H), atresia of the choanae (A), retardation of growth (R), genital hypoplasia (G), and ear abnormalities (E). The clinical phenotype involves a wide spectrum of CHDs, from mild malformations like PDA to more severe phenotypes like TOF, with conotruncal and atrioventricular septal defects being overrepresented. Mutations in the ATP-dependent chromatin modifier chromodomain helicase DNA-binding protein 7 gene (CHD7) have been identified as a major cause of CHARGE syndrome [96, 97]. With some studies suggesting an increase in congenital heart disease in patients with CHD7 pathogenic variants.
Indeed, CHD7 pathogenic mutations usually disturb the chromatin modifying activity, implying a possible epigenetic origin in CHARGE syndrome, which could be explained by the role that CHD7 plays in significant developmental pathways such as BMP [94] and altering cell migration and the development of cell lineages like the cardiac mesoderm [97]. Nevertheless, many details about its role in CHARGE syndrome remain to be elucidated, including in the context of CHD and the apparent bias towards some serious phenotypes, and understanding this topic well requires further research. The syndrome mentioned is a specific example of the participation and importance of epigenetic mechanisms in CHD.
- ATP-dependent chromatin modification is required to be described and discussed in subsection 2.
Answer: Thanks to the reviewer for this relevant observation. Next, we included information for this important issue in the section 7.
Please check LINES 505 – 533.
- ATP-Dependent Chromatin Remodeling
ATP-dependent chromatin remodeling complexes perform essential functions in the regulation of gene expression by modifying organization, structure, and accessibility of chromatin. In the nucleus of eukaryotic cells, the processes of DNA metabolism, including mechanisms of transcription, replication and DNA repair, occur from DNA packaged into nucleosomes and chromatin structures. Chromatin can be condensed, making it difficult for proteins to access it. For this, cells use enzyme complexes called chromatin remodeling factors (CRFs), which act by catalyzing the ATP-dependent restructuring and repositioning of nucleosomes [93]. These CRFs can modify the position of nucleosomes in regulatory regions on chromatin are critical for normal gene regulation, because tight interactions between nucleosomes and DNA can prevent the association of DNA with transcription factors and the core transcription machinery [94].
Chromatin remodelers are multi-subunit complexes that share a common SF2 ATPase helicase domain in their catalytic subunit that by using the energy obtained from ATP hydrolysis, they could reposition nucleosomes in chromatin, thus modifying their accessibility and the specific binding to the genome and the catalytic activity of these complexes are mediated through the different associated members of the ATPase complex. There are four main families of chromatin remodeling complexes according to their protein similarity and domain structure CHD (chromodomain-helicase-DNA binding) found in mice, SWI/SNF (switch/sucrose-non-fermenting) initially identified in prokaryotes and yeast, ISWI (imitation switch) identified in Drosophila and INO80 (inositol-requiring 80) discovered in yeast [95,96].
Therefore, the integration of different assessments such as genetic, biochemical, structural, single molecule, new omics, and biophysical has provided new insight of epigenetic modulation through chromatin remodeling complexes. Furthermore, it is necessary to emphasize continued precision research to discover new drugs to treat diseases involving defects in chromatin remodeling [96]. Given the above, it has been shown that ATP-dependent chromatin remodelers are highly relevant in developmental processes and are involved in various diseases, for instance, CHARGE syndrome.
